# Age-Related Declines in Health and Fitness among Law Enforcement Officers Compared to Population Norms

**DOI:** 10.3390/healthcare12070714

**Published:** 2024-03-24

**Authors:** Katherine A. Frick, Philip J. Agostinelli, Julia F. Swinford, Mick E. Harris, C. Brooks Mobley, JoEllen Sefton

**Affiliations:** 1Warrior Research Center, Auburn University, 301 Wire Rd, Auburn, AL 38632, USA; kaf0067@auburn.edu (K.A.F.); pja0007@auburn.edu (P.J.A.); 2School of Kinesiology, Auburn University, 301 Wire Rd, Auburn, AL 38632, USA; jfs0025@auburn.edu (J.F.S.); meh0218@auburn.edu (M.E.H.); 3Nutrabolt Applied and Molecular Physiology Laboratory, 301 Wire Rd, Auburn, AL 38632, USA; moblecb@auburn.edu

**Keywords:** ACSM guidelines, physical fitness, public safety, assessment, health

## Abstract

Physical fitness is mandatory for public safety officers. Police officers experience elevated levels of cardiovascular disease and associated risks making fitness a peak concern. Officers often have more marked fitness level decreases with aging compared to the general population. This cross-sectional study investigated the cardiovascular health, muscular strength/endurance, and mobility of officers in a medium-sized police department (N = 83); (4 females, 79 males), age (36.82 ± 10 years), height (179.02 ± 7.7 cm), body mass (95.1 ± 16.29 kg) compared to American College of Sports Medicine (ACSM) guidelines. The findings revealed that police officers begin their careers with above average strength but demonstrate greater declines with age than the general population. Officers also demonstrated cardiovascular fitness below ACSM standards and significant decreases with aging compared to the general population. Body fat percentages (*p* = 0.003) and BMI (*p* = 0.028) surpassed recommendations, with higher than normal increases with age. Maximum vertical jump decreased as officers age (*p* = 0.004). These findings support the implementation of a targeted physical fitness regimen and the resources for a program designed to improve current health and fitness, reduce the greater than expected decreases with aging, and aim to optimize occupational performance and the safeguarding of the lifelong health and well-being of officers.

## 1. Introduction

Police officers operate in high-stress and often fast-paced environments, rendering their profession one of the most physically and psychologically demanding occupations [1,2,3,4]. Law enforcement officers also experience elevated mortality and injury rates compared to many other vocations. Numerous countries report police force injury rates three times higher than other occupations [5,6,7,8]. This is primarily attributable to the demanding nature and structure of their work [7,8,9,10]. Police officers are expected to maintain a level of physical fitness that exceeds the average citizen in order to effectively carry out their duties and protect the community [11]. Many locations in the United States require passing a physical fitness test during police academy training but, once graduated, officers are not held to the same physical standard. In addition to high injury rates, officers frequently contend with health issues [1,5,7] such as obesity, metabolic disease, and an elevated risk of cardiovascular disease [4,12,13]. These health challenges can be directly attributed to job demands such as shift work, inadequate sleep, heightened stress levels, extended vehicle patrols, reduced manpower, and obligatory sedentary tasks [1,3]. Surprisingly, 80–90% of officers’ daily duties involve sedentary tasks such as paperwork and routine vehicle patrols [8]. Research also suggests healthy police officers help alleviate the city’s financial burden and enhance the quality of services provided [8]. It is essential that police officers are healthy, fit, and able to carry out their duties, thereby ensuring the safety of both the community and their fellow officers. 

The inherent risks associated with police work are widely understood [14,15,16]. Unfortunately, the repercussions on physical fitness and well-being are frequently overlooked [14,15,16]. In addition to long hours and constantly changing time demands, police officers often complete shift work changing from working day hours to night hours disrupting their circadian rhythm and sleep hygiene [16,17]. Positive health and fitness measures have been linked to decreased absenteeism which is critical in police departments. The previous literature has shown absenteeism to cost as much as $25 billion a year [14]. The need to address these issues is underscored by the elevated mortality rates experienced by police officers [18]. Factors such as cardiovascular health, stress management, physical fitness, and well-being have essential roles in determining the longevity and quality of life for those dedicated to public safety [2,5,6,7]. Police departments in smaller towns and cities encounter parallel challenges to those in larger urban centers but with fewer resources to tackle these issues. The impact of these challenges on the health and fitness of police officers in these smaller departments has yet to be evaluated.

A police department in a small, rapidly expanding city sought to establish a collaborative partnership with our academic laboratory. The objective was to address the multifaceted challenges that impact the health of law enforcement personnel [5,8,11,12]. This study represents the initial step to enhancing the health and fitness of the officers. Previous research has assessed the performance and fitness levels of police officers compared to similar physically demanding occupations, including firefighters, CrossFit athletes, and traditional sports athletes [19,20]. The American College of Sports Medicine (ACSM) provides recommendations for physical activity standards for the general population based on current science and research [21]. To our knowledge, no research has compared police officer fitness to the ACSM recommended standards. This comparison provides an in-depth evaluation of police officer fitness and serves as a foundational reference for creating officer health and fitness goals and the development of evidence-based wellness initiatives tailored to police officers. The aim of this study was two-fold: first, to compare the fitness evaluation scores of police officers to ACSM age-matched general population recommendations and secondly, to compare the health-based fitness standards within a police department by age groups. This study provides officers with a realistic and meaningful gauge of their health and fitness. We hypothesized police officers would exhibit lower fitness levels than the ACSM guidelines. Moreover, these disparities would be accentuated as officers age to a greater degree than the general population.

## 2. Materials and Methods

### 2.1. Participants

Members of a medium-sized police department (N = 83/165 (50.3% of LEO population)), (4 females, 79 males), age (36.82 ± 10 years), height (179.02 ± 7.7 cm), body mass (95.1 ± 16.29 kg) volunteered for this study. Each participant read and signed the informed consent approved by the Auburn University institutional review board. The officers completed the Physical Activity Readiness Questionnaire (PAR-Q) [21]. The participants were allowed to complete the assessment if they were deemed low or no risk on the PAR-Q or obtained a medical waiver stating that they were allowed to participate in the assessment. Nineteen of the 83 participants required physician approval prior to completing the assessments. The participants completed one testing session including body composition, sub-maximal volume of oxygen consumption (VO_2_), YMCA bench press, forearm plank, functional movement screen (FMS) pain provocation, functional movement, grip strength, vertical jump, and sit-and-reach test.

### 2.2. Body Composition Assessment

Height and weight were assessed using a Seca scale and stadiometer (SECA, Hamburg, Germany). Body composition was measured using dual energy X-ray absorptiometry (DEXA, Lunar iDXA, GE Healthcare, Atlanta, GA, USA) utilizing the pre-programmed Lunar iDXA full-body settings. The DEXA readings were used to report body mass index (BMI) and body fat percentage (BF%) [22]. 

### 2.3. Cardiovascular Assessment

The participants chose one of three submaximal (graded) cardiovascular test (GXT) options: the Gerkin treadmill test, the YMCA cycle ergometer test, or the Queen’s College step test [21,23]. The majority of the officers chose to complete the Gerkin treadmill test (60/83; 72.3%). The Gerkin treadmill test is an accepted form of VO_2_ testing predominantly for the tactical athlete population [23], encompassing a stepwise assessment that increases speed and inclines for each step of the protocol. Termination of the test occurs when the 85% heart rate maximum is reached. The YMCA cycle ergometer test was the next most frequently selected assessment (17/83; 20.5%). The requirements of the YMCA cycle ergometer test are based on the heart rate response of the participant while cycling through progressing submaximal workloads. The least selected assessment was the Queen’s College step test (6/83; 7.2%). The Queens College step test is a three-minute assessment involving stepping up and down a 41.3 cm step at a cadence of 24 steps/minute in males and 22 steps/min in females. Heart rate was evaluated at the end of the test or when the participant could no longer maintain cadence. The GXT options were provided due to the variation in age, fitness, and past injuries. The participants’ 85% of maximum heart rate was calculated using the Tanaka formula [21]. VO_2max_ was extrapolated by standard mathematical guidelines for each test [21] and compared to general population norms from ACSM guidelines [21]. Each participant was fitted with an EQ02+LifeMonitor (Equivital EQ02, Hidalgo, Cambridge, UK) using Black Ghost software (Black Ghost Version 6.3.26.2667. Hidalgo, UK) to monitor heart rate during the test. The estimated VO_2_ of the officers was compared to the normative ACSM guidelines for the general population based on biological sex and age [21]. 

### 2.4. Muscular Fitness Assessment

Muscular fitness was assessed using the YMCA bench press test. The set weight was 90 lbs. for males and 35 lbs. for females [21]. The participants completed as many repetitions as possible (up to 45) to the beat of a metronome set to 60 beats/minute. The assessment was terminated if the participants reached volitional fatigue, due to safety concerns (e.g., instability), if they were unable to maintain movement with the metronome beat, or if they reached 45 repetitions. The completed repetitions were compared to normative charts for the general population based off biological sex and age [21]. 

### 2.5. Muscular Endurance Assessment

Muscular endurance was evaluated via a 60 s forearm plank. Participants were required to keep the body in a generally straight line from torso to feet, be propped up on their toes no less than approximately 4–5” apart; place forearms below the shoulders while being flat on the ground; and with hands not touching or interlocked and staying approximately a fist’s width apart [24]. The participants were allowed to adjust within the required criteria for comfort. The assessment was terminated if the participant had to be corrected for proper positioning more than once (e.g., bringing the hips up or down to maintain a straight line), they had reached volitional fatigue, or after 60 s. The results in seconds from the muscular endurance plank test were compared against a normative general population chart for benchmarks [24]. Based on previous research [24] a 60 s plank that is deemed as “average” was utilized for the assessment. A 60 s forearm plank was employed rather than a maximum exertion timed event due to the extensive battery of assessments being conducted on a single day and the concern of fatigue interfering with test results. 

### 2.6. Power Generation Assessment 

Lower body power generation was assessed by completing two maximum vertical jumps [25] on the Leonardo Mechanography platform (Leonardo Mechanograph, Novotec Medical GmbH, Pforzheim, Germany). The participants stood on the Leonardo platform with their feet on the appropriately labeled force plates, holding as still as possible, then squatting down into a countermovement, jumping as high as possible, landing on both feet, and standing as still as possible again. Participants were allowed one familiarization attempt and then completed the two maximum vertical jumps. The highest jump was utilized for analysis. The height of the jump in centimeters and the vertical jump lower body power in watts per kilogram were recorded. 

### 2.7. Grip Strength Assessment

Grip strength [21] was evaluated using the Leonardo hand grip dynamometer. The participants were instructed to keep their upper arm in line with the torso and flex the elbow 90 degrees parallel with the floor. On the cue of “Squeeze”, the participants applied maximal force to squeeze the dynamometer for a count of five seconds then were cued to “Relax” upon test completion. Grip strength was assessed three times in each hand (alternating sides each time). The maximum grip strength from both left and right hand were noted and combined for a total grip strength sum reported in kilograms.

### 2.8. Functional Movement Assessment

Three functional movement screen (FMS) pain provocation tests [26] (shoulder clearing, active scapular stability, spinal extension, prone press up, spinal flexion, and quadruped posterior rock) were used to assess injury risk and provide insight into possible kinetic dysfunction. The awarded score was a zero if the participant experienced no pain throughout the movement, and a one if pain was present during the movement. The FMS modified pushup was used to assess trunk stability and the overhead squat to assess full body kinetic function. The scoring for each of the tests is based on the FMS ordinal scoring method [26]. The movement was scored as a three if completed fully without compensation or deviation from the movement pattern, a two if the participant demonstrated mild compensation or movement inadequacy, and a one if the participant was not able to complete the movement or had significant compensation or inadequacy. The researchers used the FMS script and scoring system [26]. The researchers were familiarized with the FMS protocol and grading was agreed upon by the researchers. The pain provocation and functional movement screen assessments were graded using the FMS scoring system detailed above [26].

### 2.9. Flexibility Assessment

Lower back and hamstring flexibility were tested using the sit-and-reach assessment [21]. The participants sat on the ground without shoes, placing their feet against the base of the box. They were instructed to stack their hands with palms down and push the bracket as far as possible with a single fluid movement, holding the stretched position for two seconds. The participants were allowed up to two practice trials and then completed three recorded trials. The length that the bracket was moved was recorded in centimeters. The maximum length was compared to normative data [21] for the general population based on age and biological sex. 

### 2.10. Statistical Analysis

Statistical analysis was completed using the R (R Core Team, 2020), R Studio (RStudio Team, 2023) version 4.2.2, and psych, lattice, ggplot2, dplyr, and tidyr packages. The normality of the data was assessed by a Shapiro–Wilkes analysis. Data that did not conform to normality by the Shapiro–Wilkes analysis were visually inspected with a Q-Q plot to determine normality. Means, minimums, maximums, and standard deviations were calculated for age groups for each assessment except the pain provocation measures and functional fitness measures, which used an ordinal rating system. An analysis of variance (ANOVA) was conducted to assess age group results compared to ACSM general population guidelines [20]. Post hoc analysis was completed to determine pairwise comparisons between age groups utilizing a Bonferroni adjusted *p*-value. [27] Kruskal–Wallace chi-square testing was conducted to investigate differences between age groups across the assessments regardless of population norms. One-sample t-tests were conducted to compare properly the age groups to ACSM standards. 

## 3. Results

The demographics of the police officers who volunteered for this study are outlined in Table 1. Group comparisons are for police officers compared to the ACSM general population standards for the officers’ specific age range (20–29 years old, n = 23; 30–39 years old, n = 37; 40+ years old, n = 29) (Table 2). Bivariate correlations were computed between age and each physical fitness measure during preliminary analysis which did not yield any significant effects.

### 3.1. Body Composition

BMI was significantly different by age group (F(2,80) = 3.742, *p* = 0.028, η^2^ = 0.09) (Figure 1A). Post hoc analysis revealed significant differences between officers’ age groups for the 20–29 years and 40+ years groups (*p* = 0.03). BMI was also significantly higher for each of the three age groups: the 20–29 years *t*(82)= 13.04, *p* < 0.001, the 30–39 years *t*(82) = 16.85, *p* < 0.001, and the 40+ years *t*(82)= 20.53, *p* < 0.001, compared to ACSM standards. The measured body fat percentage (Figure 1C) was significantly different between the age groups (F(2,80) = 6.2406, *p* = 0.003, η^2^ = 0.13). Post hoc analysis indicated a significant difference in body fat percentage between 20–29 years and 30–39 years (*p* = 0.031) and between 20–29 years and 40+ years (*p* = 0.003). The body mass index was also found to be significantly significant between the age groups χ^2^ (2, N = 83) = 8.281, *p* = 0.016) (Figure 1A). Body fat was also significantly higher for each of the three age groups 20–29 years *t*(82) = 7.93, *p* < 0.001, the 30–39 years *t*(82)= 11.55, *p* < 0.001, and the 40+ years *t*(82) = 13.53, *p* < 0.001. 

### 3.2. Cardiovascular

The estimated VO_2_ maximum was below ACSM guidelines (Figure 2) for all age groups: the 20–29 years *t*(82)= −13.79 *p* < 0.001, the 30–39 years *t*(82)= −12.47, *p* < 0.001, and the 40+ years *t*(82)= −5.16, *p* < 0.001 (Table 2). 

### 3.3. Muscular Strength and Endurance 

The age groups all tested statistically above general norms for the YMCA bench with the 20–29 years *t*(82)= 12.69, *p* < 0.001, the 30–39 years *t*(82) = 11.18, *p* < 0.001, and the 40+ years *t*(82)= 10.13, *p* < 0.001, (Table 2). The YMCA bench press was also significantly different by age (F(2,80) = 5.4559, *p* = 0.006, η^2^= 0.12) (Figure 1G). Post hoc analysis indicated a significant difference for the YMCA bench press between the 20–29 years and the 40+ years (*p* = 0.014) and between the 30–39 years and the 40+ years (*p* = 0.020). Vertical jump height was found to be significantly above the general population mean for the two younger age groups, the 20–29 years *t*(82)= 9.76, *p* < 0.001, and the 30–39 years *t*(82)= 3.87, *p* < 0.001 (Table 2). However, no significant difference was found between general population standards and the 40+ years *t*(82) = −1.61, *p* = 0.11. Statistical analysis did show a difference between the assessed age groups as well χ^2^(2, N = 83) = 10.827, *p* = 0.004 (Figure 1B). 

## 4. Discussion

This study examined the health-based physical fitness of police officers and compared the participants to the ACSM guidelines for the general population and by age group. Our investigation highlights a concerning trend: a continual decrease in overall health and fitness levels among police officers in general and decreases with age which are over and above decreases commonly found in the general population. Understanding and mitigating the unique fitness challenges faced by police departments, especially in smaller cities with fewer resources, is essential to officer well-being and contributes to the overall officer performance and longevity of law enforcement careers. This is especially important in view of widespread officer staffing shortages and increasing officer recruitment challenges. 

### 4.1. Cardiovascular 

The most concerning finding was that 82% of the youngest officers (20–29 years) scored in the very poor or poor VO_2_ Max category compared to the ACSM guidelines [21]. Cardiovascular efficiency is expected to be higher for law enforcement officers than the general population [1,7,18]. Previous research is conflicting, with research revealing a deficiency in cardiovascular fitness for law enforcement officers globally [4], while other older studies indicating above average cardiovascular fitness levels [28]. The cardiovascular assessment findings for the officers in the current study contradict the older research finding police officers with higher VO_2_ max than the general population and is similar to more modern studies for general police population. Previous research conducted over 30 years ago found an average VO_2_ max of 42.6 mL/kg/min in the general police population [28], while a special operations unit averaged 51.06 mL/kg/min [29]. Although older research has found above average cardiovascular measures within the police population, the substandard cardiovascular fitness found with our sample population demonstrates a clinically significant downward trend. This finding could suggest a heightened risk for cardiovascular disease [4,11] which contributes to comorbidities associated with extended periods of hyper-alertness due to shift work and prolonged working hours [17,30,31]. It is not uncommon to see individuals testing below the average recommended norms in the general population with the current issues of above average obesity rates and low physical fitness [32]; however, general fitness standards for police officers are higher due to job requirements [1,7,18]. Cardiovascular fitness is a fundamental job requirement, essential for successful completion of occupational tasks. The rate of decline in cardiovascular endurance observed as law enforcement officers age is especially concerning, particularly when compared to the anticipated decline found in the general population (Figure 2). The 20–29 age group was 13.6% below the ACSM guidelines, the 30–39 age group was 21.2% below, and the oldest group of 40+ year olds were 25% below (Table 2, Figure 1D). This indicates a more rapid decline in cardiovascular fitness with older police officers compared to the expected decline found in the general population as represented in the ACSM cardiovascular guidelines. A 2017 study grouped police officers by age and found a slight decline with age that is physiologically expected [33]. Despite the small decline in VO_2_ max, the police officers assessed maintained cardiovascular capacity over the expected general population norms with 20–29-year-olds testing at 44.9 mL/kg/min, 30–39-year-olds at 40.5 mL/kg/min, and 40–49-year-olds at 37.5 mL/kg/min [33]. The low levels of cardiovascular fitness found in several studies may be attributed to prolonged sedentary job responsibilities coupled with high stress levels, inadequate nutrition, and chronic sleep deprivation experienced over the course of their careers, and seem to be declining over time [1,6,7,17].

### 4.2. Body Composition

The results also revealed a prevalence of officers being overweight and above the recommended body fat percentage. This tendency exacerbated with age, with potential implications for both health and job performance. It is well documented that individuals classified as overweight or with a high body fat percentage face a higher risk for metabolic disease, a factor that poses a significant concern for the health and well-being of officers [34]. While body fat percentage generally increases as people age, the concerning aspects of the results of this study show an above average body fat percentage gain in the assessed officers compared to general population. The previous literature has also indicated a direct correlation between increased body fat percentage and decreased overall fitness according to Cooper institute standards [35]. Excess body fat also compromises the capacity to effectively carry out the demanding duties inherent in law enforcement [36], and obese workers are 23–43% more likely to experience injuries compared to workers within normal weight ranges [37]. Clearly, higher levels of overweight officers may contribute to decreased health, fitness, and performance and increased city medical costs. Programs and resources to address these issues should be put in place to assist officers in improving their overall health and wellness. 

### 4.3. Muscular Strength and Endurance 

The strength and muscular endurance of law enforcement officers is critical for job performance and is often expected to surpass general population norms [1,8,11]. Officers in the current study performed well above established standards in the YMCA bench test. Overall, the tested population completed a mean of 33 bench press repetitions (Table 2, Figure 1G). The 20–29 age group scored 2% above the ACSM guidelines, the 30–39 age group scored 4% above and the 40+ age group scored 12% above the YMCA bench press norms for the general population. Our findings were similar to those in the previous literature exploring other tactical populations, such as firefighters, who maintained similar scores in the “well above average” and “above average” categories with a mean score of 43.8 repetitions [38], and a previous study assessing firefighters with a mean score of 30.4 repetitions [39]. Notably, a considerable decline in strength relative to the general population’s normative values was observed with increasing officer age. These findings suggest that although officers generally commence their careers with strength levels above the general population average, their strength tends to fall as they age (Figure 1G).

Muscular endurance was assessed using the forearm plank, which primarily targets core endurance but also reflects overall muscular endurance. A 60 s forearm plank was used to assess a normative value [21] instead of completing a forearm plank for maximum time due to the number of assessments being completed in a single session and the fear of fatigue impacting subsequent test results. Officers tested on their time off, and all assessments had to be completed in one visit in order to reduce the burden on officer downtime. The results indicated that participants with higher body fat percentages were still able to complete the 60 s forearm plank. Our findings are contrary to previous research indicating a correlation between obesity and lower forearm plank times in the general population [23]. This divergence is likely attributed to the elevated levels of muscular strength observed, especially among younger participants. Whereas the younger officers have higher measured strength compared to the general population, it is likely that the muscular endurance follows in a similar manner. As our participants only completed a 60 s plank, instead of a maximum effort time plank we are unable to explain this finding definitively. 

A potential association may exist between plank scores and the equipment officers wear while on duty. The protective equipment worn on the torso and the duty belt may influence core function. The duty belt often causes an anterior pelvic tilt, which results in heightened muscle activity and prolonged lengthened isometric muscle contraction of the gluteus maximus, gluteus medius, transverse abdominus, rectus abdominus, and internal oblique. The pelvic positioning, particularly the anterior musculature of the core, is likely an adaptation to the load carriage around the waist. Officers seem to present with lower crossed syndrome [40], which puts the lower back at risk due to increased lumbar extension and decreased hip flexion [40]. It has been established that the typical daily protective gear and duty belt contribute to officer lower back pain [37]. The anterior tilt of the pelvis employed to compensate for the 22 kg (15 lbs) or higher load of the belt and equipment is associated with the overactivity in the anterior core musculature and may contribute to the static contraction measured during the forearm plank [37,40]. It is worth noting that this differs from the dynamic stabilization required for functional movements and core stability discussed below.

### 4.4. Functional Movements and Flexibility

Functional movement and flexibility were assessed using the sit and reach test, modified pushup, and overhead squat (Figure 3). Functional movements and movement patterns were examined rather than focusing on isolated joint range of motion, with the goal of understanding the effectiveness of the kinetic chain. There was a numerical difference, but no statistical trend showing a decrease in the FMS modified pushup score with age. Both the forearm plank and modified push up evaluate core strength, but the findings differ from the two assessments. The forearm plank assessment is a static measure of core strength, while the modified pushup evaluates the stability of the core throughout the functional movement pattern [21,26]. These findings indicate that officer core strength overall may decrease slightly over time, and functionality of the stabilization of the core is decreasing with the age. Comparing the results of these two assessments may suggest the isometric overactive musculature of the core and abdominal muscles mentioned previously may be due to adjustments to the weight of the protective equipment and duty belt: rather than true functional strength. Lower back pain, a common officer complaint, can be exacerbated with the lack of functional stabilization ability of the core compounded with tonic overactive anterior core musculature [24,40].

The sit-and-reach results revealed lower than desired flexibility in the lower back and hamstrings, particularly among male officers aged 20–39. All age groups were below the recommended ACSM general population guidelines; with the 20–29 age group 16% below, the 30–39 age group 16.9% below, and the 40+ age group 12.4% below ACSM general population guidelines for the sit-and-reach assessment. This finding highlights a prevalent issue of limited hamstring and lower back flexibility in the law enforcement population [8]. Extended sitting from long patrol times and deskwork and strain from protective gear and duty belts, as discussed above, likely contribute to widespread occurrences of lower back pain in the law enforcement community [5,8,37].

There was a trend showing a decrease in the FMS modified pushup score with age. As the modified pushup test is a measurement of core stability and functionality rather than a muscular strength or endurance test, the noted decrease with age illustrates a decrease in maintaining trunk stability as the officers age. This is concerning as trunk stability is imperative for individuals to provide proper energy transference from the lower to upper body as well as protecting from back injuries [40]. It is interesting to note that these findings differ with the forearm plank assessment, as both are tests evaluating core strength. The two tests differ in what is evaluated, with the forearm plank assessment being a static measure of core strength, while the modified pushup evaluates the stability of the core throughout the functional movement pattern [21,26]. These findings indicate again that core strength may decrease slightly over time in this population, as we see in the forearm plank, but the functionality of core stabilization noticeably decreases with officer age, as represented with the modified pushup assessment. Comparing the results of these two assessments may also support the theory that tonic overactivity of the musculature of the abdominal muscles may be due to adaptations from the weight of the protective equipment and duty belt rather than true functional strength.

### 4.5. Vertical Jump

The results of the vertical jump indicated differences between the tested age groups (Figure 1B). It is to be expected that, as officers age, lower body power generation, measured by way of vertical jump, should decrease. However, a trend was apparent that the older officers fell away from the average jump height more rapidly than to be expected [21]. The vertical jump is a well-known method to investigate lower leg power [21]. Previous tactical-specific research has indicated that vertical jump performance can be directly related to specific police task performance [41,42] and even as an illness or musculoskeletal injury predictor [43]. The quickly falling lower body power generation of the older officers indicates concern not only for increased musculoskeletal injury, but also concern for continued capability of job performance tasks.

### 4.6. Grip Strength

Our study found a continual average sum grip strength for our sample population that was above the ACSM general population standards (Table 2) (Figure 1G). This finding is interesting to note as grip strength is often compared with overall health [44] and fitness [45], as well as cardiovascular health [44,45]. Previous research has found a relationship between handgrip strength and police task performance [46]. We did not assess the relationship of any measures to police task performance in this health-based study. Comparing our findings to the previous literature findings suggests that these officers would test well on specified police task performance measures based on their grip strength measures.

The limitations of this study include the imbalance in the representation of sexes within the police department. The national average of female to male officers is approximately 13.3% female to 86.7% male [36], whereas only 4.8% of our study participants were female officers. Additionally, the challenge of screening a department of police officers while ensuring coverage of duties meant we were unable to control physical activity prior to the assessments. Some participants reported experiencing muscle soreness or fatigue due to prior workouts or job-related demands during testing. Furthermore, factors such as nutrition, consumption of caffeine, and nicotine were not standardized during the study. The lack of control over stimulant intake within 24 h before testing could potentially impact the measured outcomes, particularly in the case of submaximal VO_2_ measures, which rely on calculations based on 85% of maximum heart rate. The previous literature has found that healthier individuals tend to volunteer for health-based studies. This is known as the “Healthy Worker Effect” and has the potential to impact research findings [32]. As all our participants were volunteers from a single police department, this effect may have impacted our findings resulting in estimations of higher levels of fitness than actually exist in the department.

## 5. Conclusions

These findings revealed that although law enforcement officers typically commence their careers meeting or even surpassing general population fitness standards, their fitness levels diminish more rapidly than expected as they age, ultimately falling notably below ACSM guidelines. These results highlight the importance of implementing a comprehensive wellness and fitness program specifically tailored to the unique demands and available resources of a small city police force. Fitness and wellness initiatives, along with adequate resources (time, equipment, expertise, and education), are vital to mitigating potential health risks and enhancing the long-term quality of life of law enforcement officers, both during their active service and post-retirement.

## Figures and Tables

**Figure 1 healthcare-12-00714-f001:**
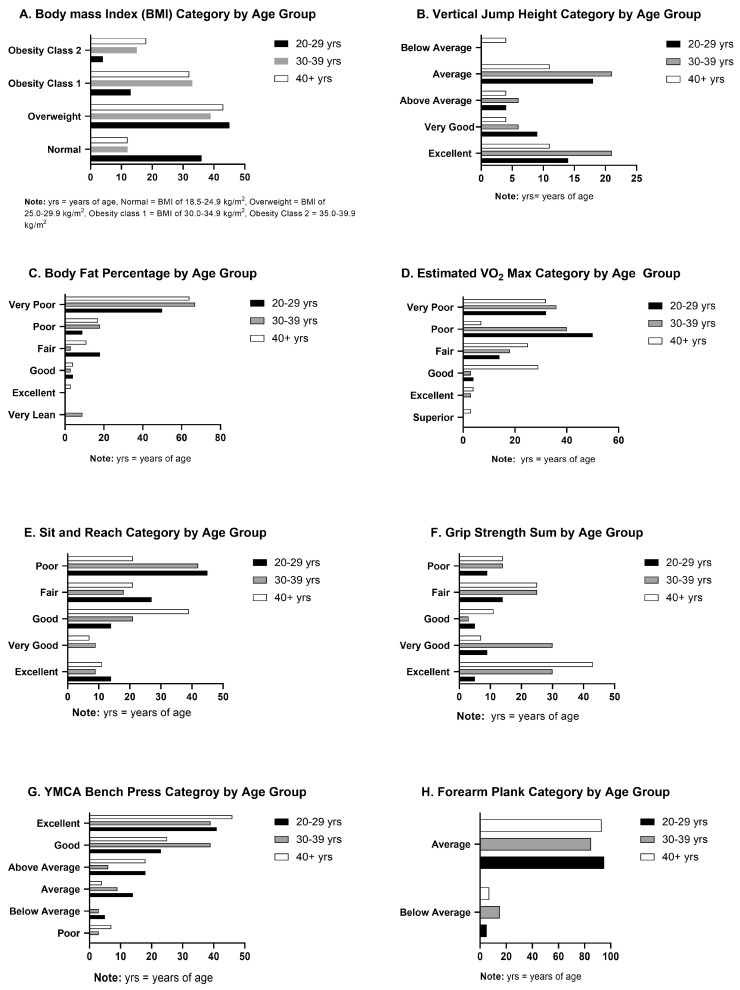
Assessment Results by Age Group Percentage.

**Figure 2 healthcare-12-00714-f002:**
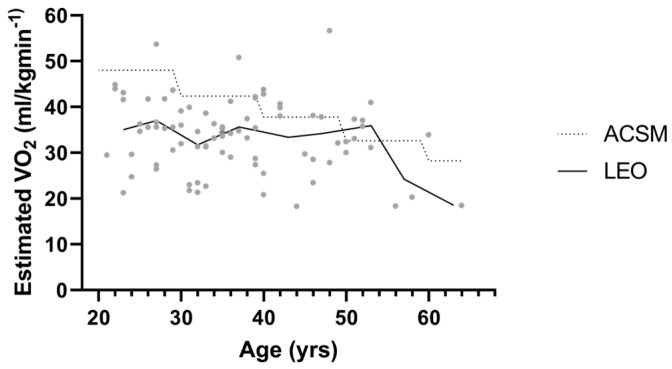
Estimated VO2 Maximum by Age Group vs. General Population. LEO = police department, ACSM = ACSM general population guidelines, yrs = years of age.

**Figure 3 healthcare-12-00714-f003:**
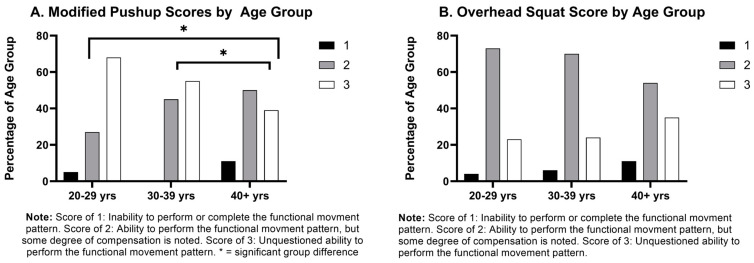
Movement Assessment Results by Percentage of Age Groups.

**Table 1 healthcare-12-00714-t001:** Descriptive Statistics.

Variable	Mean ± SD	Minimum	Maximum
Age (yrs)	36.82 ± 10.00	21.00	64.00
Height (cm)	179.02 ± 7.70	162.50	203.20
Weight (kg)	95.10 ± 16.26	58.42	131.45
Body Fat Percentage	27.73 ± 7.09	8.80	40.50

Note: yrs = years; cm = centimeters; kg = kilograms.

**Table 2 healthcare-12-00714-t002:** Assessment Results by Age Groups.

	LEO 20–29 yrs n = 23	ACSM 20–29 yrs	LEO 30–39 yrs n = 37	ACSM 30–39 yrs	LEO ≥ 40 yrs n = 29	ACSM ≥ 40 yrs
Estimated VO_2_ Max (mL/kg/min)	* 36.12 ± 7.85	48.00	* 33.42 ± 6.56	42.40	* 32.65 ± 9.09	37.80
YMCA Bench Press (# repetitions)	* 36.27 ± 8.81	22.00–26.00	* 35.12 ± 9.06	22.00–26.00	* 28 ± 11.69	12.00–18.00
Forearm Plank (seconds)	59.25 ± 3.54	60.00	58.09 ± 5.79	60.00	58.8 ± 4.48	60.00
Grip Strength Sum (kg)	99.15 ± 22.30	95.00–103.00	101.08 ± 19.80	95.00–103.00	* 99.86 ± 23.32	86.00–92.00
Vertical Jump (p.max/kg)	* 55.81 ± 12.22	45.72	* 50.01 ± 10.09	45.72	44.19 ± 8.64	45.72
Sit and Reach (cm)	* 26.45 ± 9.57	30.00–33.00	* 24.94 ± 9.20	28.00–32.00	* 22.77 ± 9.32	24.00–28.00

Note: Results are reported as mean ± standard deviation, * indicates significance, LEO = police department, ACSM = ACSM general population guidelines, yrs = years of age, kg = kilograms, cm= centimeters, p.max/kg = power maximum/kilograms.

## Data Availability

The data presented in this study are available upon request from the corresponding author through FigShare.

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
