# Peer review of "Age-Related Declines in Health and Fitness among Law Enforcement Officers Compared to Population Norms"

_healthcare, 2024, doi:10.3390/healthcare12070714_

Round 1
Reviewer 1 Report
Comments and Suggestions for Authors
General Comments:
The study investigated physical fitness of police officers in an urban setting. Overall the study was interesting and addresses an important issue for the health of police officers. The aims of the paper appear to be 2-fold: 1) to assess differences in measures of physical fitness due to age and 2) compare the observed fitness levels to ACSM guidelines. The first aim can be more clearly stated in the introduction where at present describes mainly aim 2. The rationale for examining physical fitness trends of police officers in a ‘small, rapidly growing city’ can be strengthened in the introduction. The reviewer does appreciate the work of the research team and service provided to police officers who participated in the study.
Specific Comments
Abstract
-The colon in the 3rd sentence isn’t needed
-Can you please add some more detail to the methods. Particularly, it is not clear if the design was longitudinal or cross-sectional.
-Quantifying some of the findings in the abstract would be beneficial
-If space permits I’d recommend noting the components of fitness were ‘health-based’ instead of performance.
Introduction
-Line 34: appears to be an extra space to start the sentence. Same on line 37. Please check the entire manuscript for consistency with number of spaces to start sentences.
-End of paragraph 1: Something else to consider is that many law enforcement agencies are finding it difficult to recruit new law enforcement officers (LEOs) and as a result are understaffed. One solution is to try to keep current LEOs as healthy as possible for career ‘longevity’.
-Line 37-39: Several studies have provided evidence that physical fitness, at start of academy, is able to predict academy graduation. Considering the common physical fitness tests to enter academies and graduate many LEOs start careers with adequate levels of fitness.
Line 52-53: Can you elaborate on the studies you cite here.
Paragraph 3 – Some more context as to the importance of investigating physical fitness in a ‘small, expanding city’ would strengthen the introduction.
Line 63-65: Can you please state the specific ACSM recommendations you will compare to. This may not be common knowledge for all readers.
Methods
Recruiting – It is mentioned that the LEO ‘volunteered’ for the study. So although half of the department participated there could be some bias in the findings. Lockie and colleagues have noted a ‘Healthy Worker Effect’ where it is common when participation is voluntary the healthier, in this case fitter, workers are the ones who participate. Please consider this as a limitation and how it may influence your results. My lab has run into a similar issue and unfortunately I have no solution but is a limitation worth acknowledging.
Chowdhury R, Shah D, Payal AR. Healthy Worker Effect Phenomenon: Revisited with Emphasis on Statistical Methods - A Review. Indian J Occup Environ Med. 2017;21(1):2–8.
Muscular endurance – can you please clarify if this was score as pass/fail. Why not do a maximal effort as opposed to 60 s cutoff?
CMJ – where any warm-up or practice attempts given?
Statistical analyses – please comment on the normality of data and whether outliers were checked for. Also, what where the group sizes for the 3 age groups? The normality and sample size will help to clarify whether parametric statistics were appropriate. Please also report effect sizes in the results.
-Categorizing into age groups, while common, in the literature does create a potential ‘loss of information’. See this commentary but no need to reference:
Altman DG, Royston P. The cost of dichotomising continuous variables. BMJ. 2006 May 6;332(7549):1080.
Computing and reporting bivariate correlations between age with each measure of fitness would be of value. This could complement current analyses.
Results
-Pending any potential changes to the statistical analyses the organization could be improved. There seems to be 2 aims: 1) examining age differences and 2) comparing to ACSM guidelines.
Discussion
-A few broad comments for the discussion. First, would compare/contrast your reported values to previously published in LEO. Then the second aspect I believe needs attention is that the importance of a ‘small, rapidly growing city’ is not really discussed. That aspect could be touched upon more. I suspect it relates to resources and to support the health and wellness of LEO departments may need to be more proactive, than reactive, to growth.
-The discussion header appears to be missing.
-Paragraph 2 – is the downward trend in cardiovascular fitness unique to LEOs? Or is this a generational shift common to LEO and non-LEO populations. Similar comment for other areas of fitness.
-Line 272-276 – if exists please provide some references.
Tables
-Please add the n for each age group to table 2.
Figure 1
-Adding the individual data points to this figure would be more informative in my perspective. Right now you present the group level data but individual data points would help to understand how many individuals were below the ACSM guidelines.
Comments on the Quality of English LanguageSuggest some editing for organization and grammar.
Reviewer 2 Report
Comments and Suggestions for Authors
The many fitness-related tests are a strength of this study. However, it is difficult to understand the direction the author is trying to explain.
Publishing requires editing to improve reader understanding.
Introduction
Requests to cite other, more specific studies on police fitness.
Methods
Add a small section on research methods.
For example) 2.1. Participants / 2.2. Muscular fitness test / 2.3. Muscular fitness/ 2.4… .2.5… . 2.6.
Please add a sitation reference.
“Lower body power generation, Grip strength, Three Functional Movement Screen, Low back”
Results
Table 1. Unify decimal points and spacing.
Figure 1. Please provide the abbreviations for LEO and ACSM in the footnotes.
Please indicate statistical significance (e.g., *) for the variables in figure 1, table 2, and figures 2 and 3.
There is a lot of result information provided by figures and tables, but there is very little description. Please write about the main results.
The results section also needs to be divided into three sections, 3.1. / 3.2. / 3.3. It is recommended to organize it by .
Discussion
Where does the Discussion begin?
The authors review each test in paragraphs, but comparisons with other studies are very poor. Whether your fitness is ‘high’ or ‘low’ is a result. The author continues to repeat the results.
I hope you will make many references to other studies. It is also meaningful to refer to research on soldiers or firefighters, public occupations that require physical strength similar to that of police officers.
Ultimately, the meaning of fitness for police officers is not limited to just cardiovascular disease. Their high fitness requirements are linked to crime detection and job performance. These should be emphasized more in the discussion and referenced with other studies.
Round 2
Reviewer 1 Report
Comments and Suggestions for Authors
The authors have done well addressing the revisions from the first round. It would be fine for the authors to add a single line to the first paragraph of the introduction (line 210) along the lines: During preliminary analyses bivariate correlations were computed between age and each physical fitness measure, which did not yield any significant relationships.
Reviewer 2 Report
Comments and Suggestions for Authors
Thank you very much for your sincere revision.
In introduction citations, one to three representative citation at the end of each sentence are sufficient. There are sentences with too many citation. Example line 34: [5-10]. It is 6 references.
Comments on the Quality of English Language
I do not have any comment.
